# Hyponatremia Due to VZV-Induced SIADH in an Older Patient: Case Report and Literature Review

**DOI:** 10.3390/idr17050106

**Published:** 2025-08-30

**Authors:** Zuzanna Żak-Skryśkiewicz, Patrycja Krupińska, Carlo Bieńkowski, Przemysław Witek

**Affiliations:** 1Department of Internal Medicine, Endocrinology and Diabetes, Medical University of Warsaw, 02-091 Warsaw, Poland; przemyslaw.witek@wum.edu.pl; 2Department of Internal Medicine, Endocrinology and Diabetes, Mazovian Bródnowski Hospital, Kondratowicza 8, 03-242 Warsaw, Poland; p.adamek@brodnowski.pl; 3Department of Adults’ Infectious Diseases, Medical University of Warsaw, Wolska 37, 01-201 Warsaw, Poland; carlo.bienkowski@wum.edu.pl

**Keywords:** hyponatremia, SIADH, varicella zoster virus, electrolyte imbalance

## Abstract

Introduction: Hyponatremia is a common finding in hospitalized patients, especially the elderly. Symptoms of hyponatremia can vary depending on the concentration of sodium in serum as well as the dynamics of its escalation. Hyponatremia can have many etiologies, including medication, vomiting, or diarrhea, and central nervous system disorders, including tumors, trauma, and infections. Case report: In this case, we present a 74-year-old patient who was admitted to the Department of Internal Medicine with symptomatic, acute, and severe hyponatremia in the course of the syndrome of inappropriate antidiuretic hormone secretion due to varicella zoster virus meningoencephalitis. Clinical improvement and normalization of natremia occurred after the initiation of causal treatment. Conclusion: Given the complexity of the potential causes of hyponatremia and the variety of treatments available, it is essential to thoroughly consider the possible reasons for electrolyte abnormalities, including uncommon ones such as central nervous system infections.

## 1. Introduction

Electrolyte disturbances are a frequent challenge in clinical medicine, with hyponatremia being the most common and potentially life-threatening abnormality encountered in hospitalized patients [1,2]. In older adults (>60 years old), even mild reductions in serum sodium can precipitate significant morbidity, including confusion, falls, and prolonged hospitalization [3]. The clinical complexity increases when hyponatremia presents alongside neurological symptoms, prompting urgent investigation into possible central nervous system (CNS) causes.

Hyponatremia, defined as a serum sodium concentration ([Na+]) < 135 mmol/L, affects 15–33% of hospitalized patients worldwide, especially among the elderly [1,2,3], partly due to the higher burden of comorbidities and increased exposure to polypharmacy [3].

The etiology of hyponatremia is complex, and its clinical presentations may range from asymptomatic to life-threatening. Therefore, managing the illness successfully requires an awareness of its potential causes. Factors known to influence low sodium concentration include gastrointestinal losses, diuretic use, and systemic illnesses such as congestive heart failure and renal and liver diseases [4]. Moreover, CNS pathologies, including trauma, tumors, or infections that manifest as the syndrome of inappropriate antidiuretic hormone secretion (SIADH) and cerebral salt wasting syndrome (CSW), may also be responsible for hyponatremia [5]. Between 30–64% of CNS infections may occur with hyponatremia [6,7,8]. The most common etiologies of CNS infections include viruses, especially enteroviruses [6,9,10]. However, Herpesviridae, including herpes simplex viruses (HSV) and varicella zoster virus (VZV), may also be responsible for the disease [6,9,10,11].

VZV, the causative agent of both varicella (chickenpox) and herpes zoster (shingles), can reactivate later in life, particularly in older or immunocompromised individuals, and spread to the CNS [12]. This reactivation may lead to meningoencephalitis, which, in turn, can trigger SIADH by affecting hypothalamic or pituitary function and disrupting the regulation of antidiuretic hormone (ADH). As a result, water retention and dilutional hyponatremia ensue [13]. VZV-related SIADH is rare but clinically important, as it can be easily overlooked without a high index of suspicion.

In this paper, we report a patient with hyponatremia due to SIADH caused by VZV meningoencephalitis. This case highlights the importance of maintaining a broad differential diagnosis in elderly patients presenting with hyponatremia and altered mental status and emphasizes the need to consider VZV as a potential yet often overlooked cause of CNS infection and secondary SIADH.

## 2. Case

A 74-year-old man presented to the emergency department (ED) with non-anginal chest pain and abdominal pain not responding to nonsteroidal anti-inflammatory drugs and antispasmodic medications. His medical history included an ischemic stroke in 2020, from which he made a full recovery, and atrial fibrillation, for which he was on chronic anticoagulation with apixaban 5 mg twice daily. The physical examination revealed normal vital signs; the Glasgow Coma Score (GCS) was 15.

The electrocardiogram showed atrial flutter, no ischemic changes, and negative markers of myocardial damage. After receiving 600 mg of amiodarone intravenously (IV) and undergoing observation, the patient was discharged with a recommendation to take 40 mg of proton pump inhibitor (PPI) orally and referred to report for scheduled electrical cardioversion after three weeks. The sodium concentration measured that day was 137 mmol/L.

Two days later, the patient was admitted again to the ED due to nausea and persistent abdominal pain. Laboratory testing performed in the ED revealed a C-reactive protein (CRP) level of 3.8 mg/L (normal range 0–5 mg/L) and a sodium [Na+] level of 114 mmol/L, while glucose concentration and renal parameters were within a normal range. No abnormalities were found in the abdominal ultrasound and X-ray. Chest X-ray showed consolidations in the right lung, and the heart silhouette was normal-sized. The patient was diagnosed with severe hyponatremia and community-acquired pneumonia. Treatment included hypertonic saline (3% NaCl), spasmolytic drugs, PPI, and antibiotics. During the ED hospitalization, the patient experienced an epileptic seizure, which resolved spontaneously. A head computed tomography (CT) scan without contrast was performed, which showed no fresh ischemic or hemorrhagic lesions. There was a small focus of malacia in the white matter of the left parietal lobe. After approximately 36 h of hospitalization in the ED, no clinical or laboratory improvement was observed; therefore, the patient was admitted to the Internal Medicine department for further management.

On admission to the department, the patient was in severe condition, conscious, lethargic, and received 9 points in GCS (ocular response—2/4, verbal response—3/5, motor response—4/6). The pupil’s reaction to light was normal. Meningeal symptoms were negative. Other than that, there were no significant deviations in the physical examination, and the hydration status was clinically assessed as euvolemia.

On the next day, serum [Na+] was 115 mmol/L, plasma osmolality was 225 mOsm/kg (normal range 275–300 mOsm/kg), urine osmolality was 626 mosmol/kg (normal range 100–1200 mOsmol/kg), urinary sodium was 54 mmol/L (normal range <30 mmol/L), CRP was 32.4 mg/L (normal range 0–5 mg/L), and procalcitonin level (PCT) was 0.34 ng/mL (normal < 0.5 ng/mL). In the differential diagnosis, hypothyroidism, adrenal insufficiency, hyperlipidemia, hyperglycemia, as well as previous use of diuretics were excluded. The only abnormal parameter indicating a possible alternative cause of hyponatremia was significantly elevated 4203.2 pg/mL N-terminal prohormone brain natriuretic peptide (NT-proBNP) (normal range 0–125 pg/mL).

Despite using hypertonic solution and fluid restriction, there was no significant improvement in serum [Na+] levels or the patient’s neurological condition. Therefore, magnetic resonance imaging (MRI) of the head with diffusion-weighted imaging (DWI) and fluid-attenuated inversion recovery (FLAIR) sequences was ordered. The imaging ruled out acute ischemic stroke, and various temporal vascular lesions were described. The patient was consulted by a neurologist and was qualified for a lumbar puncture (after 24 h due to apixaban use). The cerebrospinal fluid (CSF) testing revealed a clear appearance, cytosis of 50 cells/uL (normal range 0–5/uL), granulocytes 87%, monocytes 9%, lymphocytes 4%, CSF proteins 34.0 mg/dL (normal range 15–60 mg/dL), and glucose 39 mg/dL (normal range 40–90 mg/dL) with serum glucose level 84 mg/dL. Given high inflammatory parameters, X-ray changes, and suspected meningitis, broad-spectrum empirical antibiotic therapy with ceftriaxone 2 g daily IV and ciprofloxacin 400 mg twice a day IV was administered. The outcome suggested either a viral or less probable tuberculous infection.

The patient was consulted again with a neurologist and an infectious disease specialist. Therapy with intravenous acyclovir at a dose of 3 times 10 mg/kg was initiated. Two days later, the patient developed a generalized and pruritic maculopapular rash, shown in Figure 1.

Since the implementation of acyclovir, the patient’s neurological condition improved gradually, and natremia increased.

The patient was transferred to the Infectious Diseases Unit. A lumbar puncture was performed again, and CSF testing revealed similar results. The multiplex polymerase chain reaction (PCR) of the CSF detected VZV. Current therapy with acyclovir and ceftriaxone with ciprofloxacin was continued. Further clinical and neurological improvements were observed, and after the completion of the treatment, the patient was discharged home without neurological deficits. The trend in serum sodium levels over time is depicted in Figure 2.

## 3. Discussion

Hyponatremia in clinical practice can be categorized according to the time of onset, Na+ concentration, and plasma osmolality [14], as demonstrated in Figure 3.

In the case of our patient, we observed acute and severe hyponatremia. The clinical and laboratory findings strongly suggest SIADH. SIADH is characterized by an increase in ADH secretion in the absence of normal osmotic (hypertonicity) or hemodynamic (reduced effective arterial blood volume) stimuli [15]. A high level of ADH causes increased water retention with normal sodium excretion, which results in euvolemic and hypotonic hyponatremia, urine hyperosmolality, and high urinary sodium concentration [15]. CSW may present with similar clinical and laboratory findings (including hyponatremia, urine hyperosmolality, and elevated urinary sodium excretion) [16]. However, the key differentiating factor is the patient’s volume status: SIADH is associated with an euvolemic or mildly hypervolemic profile, whereas CSW typically occurs in hypovolemic individuals [16]. Since the clinical presentations of both conditions often overlap, CSW needs to be considered in the differential diagnosis [17].

In this case, the patient was euvolemic, which supports the diagnosis of SIADH. Moreover, NT-proBNP was significantly elevated, which could suggest heart failure as a potential cause of hyponatremia [18]. Nonetheless, the absence of clinical features of congestion and the correction of electrolyte disturbances after the initiation of acyclovir—without fluid therapy or diuretic modification—indicate that SIADH, associated with a neuroinfection, should be considered the most likely cause of hyponatremia.

Not every patient with VZV meningitis will develop encephalitis, in which administration of acyclovir as empiric treatment is strongly recommended. In this case, symptoms of encephalitis were not evident. However, a high index of suspicion allowed the implementation of empiric therapy, which resulted in clinical improvement. The prognosis of VZV meningoencephalitis may vary, but full recovery may be observed [19].

## 4. Review of the Literature

### 4.1. Hyponatremia in CNS Infections

Hyponatremia is a frequent and clinically significant complication in patients with CNS infections. It may reflect disease severity and can arise due to various mechanisms, including impaired water regulation, inflammation-induced ADH release, or direct involvement of the hypothalamic–pituitary axis [20]. Its presence may also carry diagnostic value or signal the need for more intensive management, particularly in older or immunocompromised individuals [20].

Lim et al. conducted a large cross-sectional study of 184 patients with clinical evidence of CNS infection and abnormal cerebrospinal fluid composition, reporting hyponatremia in almost 40% of cases. Among the 54% of patients in whom a specific pathogen was identified, two-thirds were viruses—most commonly enteroviruses (36.4%), followed by HSV (19.2%), which was associated with the highest odds of hyponatremia, and VZV (15.2%) [6]. Additionally, Czupryna et al. reported an increased incidence of hyponatremia in tick-borne encephalitis (TBE) (44.4%) compared to other viral etiologies [21].

Other studies have supported the association between hyponatremia and specific infectious agents. In a nationwide Dutch cohort of 696 adults with community-acquired bacterial meningitis, 30% had hyponatremia on admission (6% had severe hyponatremia), although it was not associated with poor outcomes [8]. Similarly, a cross-sectional study from Bangladesh observed hyponatremia in 66% of adult bacterial meningitis cases, of which 21.2% were moderate or severe [22]. A Korean cohort study noted that patients with hyponatremia were more likely to have tuberculous meningitis compared to typical bacterial or viral infections [23]. The results of Basaran et al. suggested hyponatremia to be predictive of HSV-1 encephalitis among patients with viral encephalitis [24].

Establishing the underlying cause of a CNS infection quickly is essential to guide appropriate therapy. The role of CSF multiplex PCR testing in patients with lymphocytic meningitis is crucial for the possibility of early etiotropic treatment administration [25]. Timely recognition and etiological clarification are significant in elderly individuals, who are more vulnerable to both CNS infection and hyponatremia, and are at a greater risk of VZV reactivation [26].

### 4.2. SIADH in the Context of VZV Infection

SIADH is a well-recognized mechanism underlying hyponatremia in CNS infections. While the association between SIADH and viral meningoencephalitis has been extensively described [13,21], its occurrence specifically in the context of VZV infection is less well-documented but increasingly recognized.

VZV possesses neurotropic properties and, upon reactivation, can lead to a spectrum of CNS manifestations, including vasculitis, meningitis, encephalitis, and myelitis [27]. The resulting inflammatory response, characterized by elevated cytokine levels, along with possible direct involvement of the hypothalamus or pituitary gland, may contribute to the dysregulation of ADH secretion. While population-level data is lacking, several case reports have described hyponatremia attributed to SIADH in the context of VZV-related disease, particularly in cutaneous or disseminated forms.

However, documentation of SIADH specifically in cases of VZV meningitis or encephalitis remains limited, though a few reports support this association. Cases have been described in which patients with VZV meningoencephalitis or cerebellitis developed SIADH, including presentations without accompanying cutaneous manifestations [28,29].

Additional reports have documented SIADH in patients with localized herpes zoster. Foppiani et al. reported a case of an elderly patient with localized thoracic herpes zoster who developed severe and persistent hyponatremia due to SIADH, ultimately resulting in death [30]. Similarly, Bassi et al. described an 82-year-old patient with localized herpes zoster who presented with profound hyponatremia associated with SIADH and concurrent hypokalemia [31].

Similar complications have been observed in patients with herpes zoster ophthalmicus, including SIADH and post-herpetic neuralgia. In these cases, the resolution of neurological symptoms was paralleled by normalization of serum sodium levels, suggesting a potential involvement of central regulatory pathways governing ADH secretion [32].

VZV reactivation is particularly common in immunocompromised individuals and may present with atypical or disseminated disease. A case series from China described two allogeneic stem cell transplant recipients who developed occult disseminated zoster and hyponatremia consistent with SIADH [33]. Similarly, a 64-year-old woman receiving immunosuppressive therapy for Waldenström macroglobulinemia was diagnosed with SIADH in the setting of disseminated visceral varicella zoster infection manifesting with gastrointestinal symptoms [34].

While multiple reports associate hyponatremia with VZV-related CNS infections, few explicitly identify SIADH as the underlying mechanism. This underscores the importance of considering SIADH in the differential diagnosis, as timely recognition can influence both clinical monitoring and therapeutic decisions.

## 5. Conclusions

Since hyponatremia, especially severe hyponatremia, can manifest as excessive drowsiness, convulsions and even coma, other potential causes of a patient’s altered neurological condition may be overlooked. As the symptoms generally resolve with the gradual equilibration of sodium, the lack of full neurological improvement when normonatremia is achieved should prompt a search for other causes. Less apparent etiologies, such as SIADH coexisting with CNS infection, should also be considered, particularly given that the elderly population is at greater risk of VZV reactivation. This case illustrates a rare occurrence of SIADH secondary to VZV meningitis, in which empirical antiviral treatment for suspected CNS infection was initiated before the appearance of the characteristic rash and prior to confirmation of the viral etiology. Early recognition of CNS involvement enabled appropriate management and was followed by gradual clinical and biochemical improvement. Clinicians should remain vigilant for this uncommon but clinically relevant association to ensure timely diagnosis and optimal treatment.

## Figures and Tables

**Figure 1 idr-17-00106-f001:**
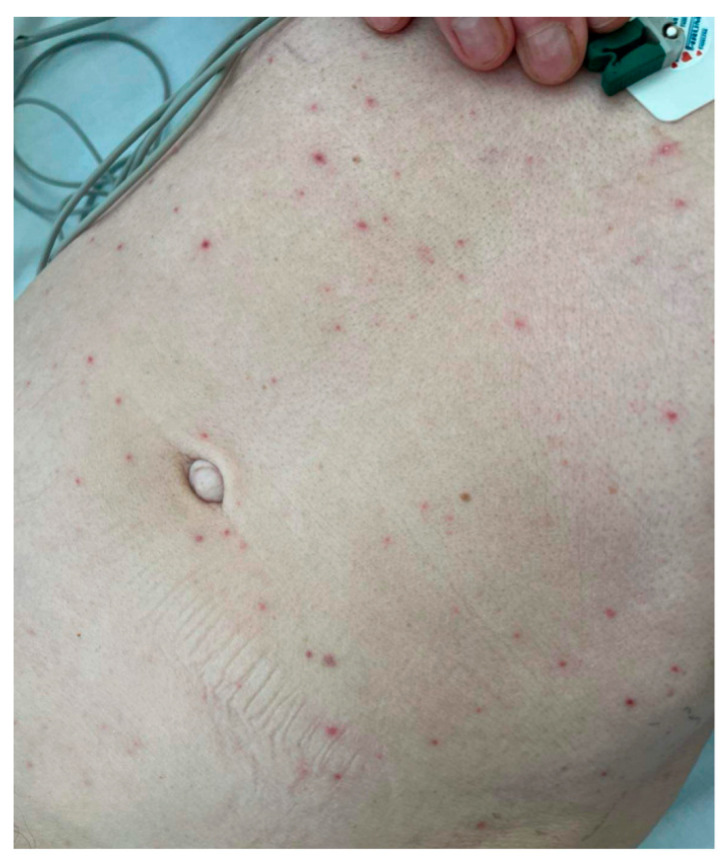
The generalized maculopapular rash patient developed on the 4th day of hospitalization.

**Figure 2 idr-17-00106-f002:**
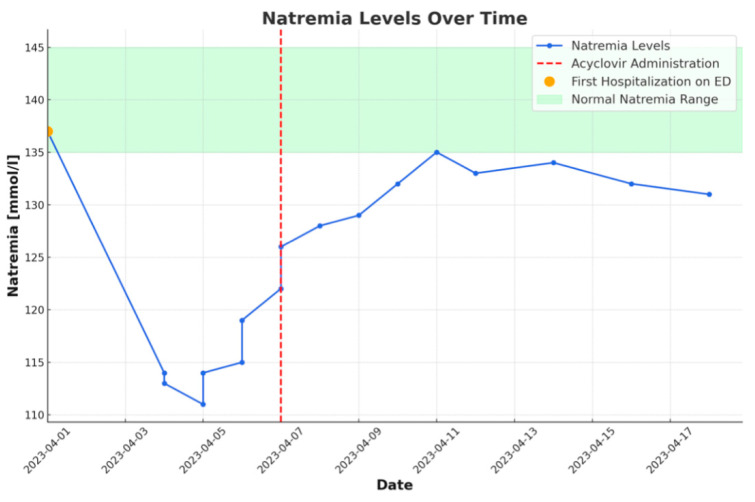
Graph showing the variation of natremia over time. Noticeable improvement after the initiation of acyclovir treatment.

**Figure 3 idr-17-00106-f003:**
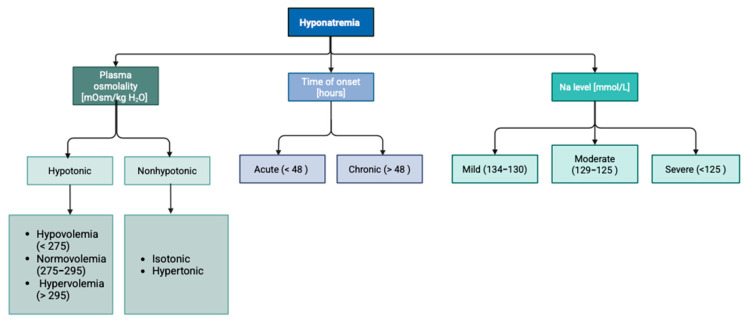
Division of hyponatremia according to plasma osmolality, duration, and sodium concentration.

## Data Availability

The data analyzed or generated during the study are available from the corresponding author on reasonable request.

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
