# Peer review of "Hyponatremia Due to VZV-Induced SIADH in an Older Patient: Case Report and Literature Review"

_2036-7449, 2025, doi:10.3390/idr17050106_

Round 1

Reviewer 1 Report

Comments and Suggestions for Authors

The case is well presented and discussed. I have only two suggestions:

  • The authors should highlight the novelty of this case and deeply discuss it compared to the available similar literature.
  • The conclusions are vague and unclear. Please rewrite this section consistently with the key messages that the authors wanted to present to readers with the presented case.
Comments on the Quality of English Language

N.a.

Author Response

Reviewer 1

The case is well presented and discussed. I have only two suggestions:

  • The authors should highlight the novelty of this case and deeply discuss it compared to the available similar literature.
  • The conclusions are vague and unclear. Please rewrite this section consistently with the key messages that the authors wanted to present to readers with the presented case.

Response: Thank you for your valuable comment regarding the Conclusions section. In response, we revised the paragraph to better highlight the clinical relevance and unique aspects of the case. We added two sentences to emphasize the rare coexistence of SIADH and VZV meningitis and clarified the clinical sequence—antiviral therapy was started empirically before rash onset and laboratory confirmation of VZV. We also refined the final sentence to provide a clearer clinical takeaway. We believe these changes address your concerns and improve the clarity of the manuscript’s conclusion.

Reviewer 2 Report

Comments and Suggestions for Authors

The authors present an interesting case of VZV-associated hyponatremia. 

Line 36: should have an “a” before “serum sodium concentration” 

Line 53-54: “ which can, in turn,” is oddly stated, would consider “which, in turn, can” 

Line 72: before the word “observation” would consider using “ undergoing”. Sounds a bit better. 

Line 72: “the recommendation” should be “a recommendation” 

Line 89: Lowercase “Department” to “department” 

Line 89: remove “a” before severe condition

Line 90: You state the patient was unresponsive? But then you give a GCS score of 9 which means minimally responsive. Would clarify this. 

Line 100: NT proBNP units incorrect. Would change to pg/ml

Lines 94 -100: I would also recommend listing the urinary sodium. This is critical information. 

Line 105: “options” should be “sequences” 

Line 108: should “apixabane” be “apixaban”

Line 108: “uptake” should be “use” 

Line 117: “presented” should be “developed” 

Line 118: Remove this: “shown in Error! Reference source not found.. “

Line 121: “presented” should be “developed” 

Line 123: remove “has” 

Line 124: remove “has”

Line 130: Would change “defects” to “deficits” 

Line 143-144: This is incorrectly stated but I know what the authors are trying to get across. The way it is written currently says that the osmolality is causing elevated levels of ADH which is not correct. Would say something like: SIADH is characterized by an increase in ADH secretion in the absence of normal osmotic (hypertonicity) or hemodynamic (reduced effective arterial blood volume) stimuli. 

Line 145: “fluid retention” should be “water retention” 

Line 156: “ionic” should be “electrolyte” 

Line 157: "accompanying" should be “associated with”

Line 159 -160: “allowing the administration of acyclovir as empiric treatment”. I am unclear what you mean by this line. Would rewrite. 

Line 172-175: I would include more explanation of the population studied: A cross sectional study of adults with CNS infection found that hyponatremia was present in 39% of cases.  Also I do not see where you got the “two-thirds of microorganisms identified from CSF were viruses”. I don't see the ⅔ reference. 

Line 187: put “a” before “CNS infection” 

Line 203: “are” should be “is”

Line 234-235: This line is a bit confusing since at first you say as symptoms improve but then you say the lack of improvement?  I would rewrite this sentence and would recommend “resolve” instead of “withdraw”.  “As the symptoms generally withdraw with the gradual equilibration of sodium, the lack of improvement, when normonatremia is achieved should prompt a search for other causes.”

Comments on the Quality of English Language

Overall the english is average. A few mistakes that can be corrected easily. 

Author Response

Reviewer 2

The authors present an interesting case of VZV-associated hyponatremia. 

Response: Thank you for acknowledging our work.

Line 36: should have an “a” before “serum sodium concentration” 

Response: Thank you for your remark. We have implemented it.

Line 53-54: “ which can, in turn,” is oddly stated, would consider “which, in turn, can” 

Response: Thank you for your remark. We have implemented it.

Line 72: before the word “observation” would consider using “ undergoing”. Sounds a bit better. 

Response: Thank you for your remark. We have implemented it.

Line 72: “the recommendation” should be “a recommendation” 

Response: Thank you for your remark. We have implemented it.

Line 89: Lowercase “Department” to “department” 

Response: Thank you for your remark. We have implemented it.

Line 89: remove “a” before severe condition

Response: Thank you for your remark. We have implemented it.

Line 90: You state the patient was unresponsive? But then you give a GCS score of 9 which means minimally responsive. Would clarify this. 

Response: Thank you for your remark. We have clarified this.

Line 100: NT proBNP units incorrect. Would change to pg/ml

Response: Thank you for your remark. We have implemented it.

Lines 94 -100: I would also recommend listing the urinary sodium. This is critical information. 

Response:  Thank you for your remark. I found the information and added it to the manuscript.

Line 105: “options” should be “sequences” 

Response: Thank you for your remark. We have implemented it.

Line 108: should “apixabane” be “apixaban”

Response: Thank you for your remark. We have implemented it.

Line 108: “uptake” should be “use” 

Response: Thank you for your remark. We have implemented it.

Line 117: “presented” should be “developed” 

Response: Thank you for your remark. We have implemented it.

Line 118: Remove this: “shown in Error! Reference source not found.. “

Response: Thank you for your remark. We have implemented it.

Line 121: “presented” should be “developed” 

Response: Thank you for your remark. We have implemented it.

Line 123: remove “has” 

Response: Thank you for your remark. We have implemented it.

Line 124: remove “has”

Response: Thank you for your remark. We have implemented it.

Line 130: Would change “defects” to “deficits” 

Response: Thank you for your remark. We have implemented it.

Line 143-144: This is incorrectly stated but I know what the authors are trying to get across. The way it is written currently says that the osmolality is causing elevated levels of ADH which is not correct. Would say something like: SIADH is characterized by an increase in ADH secretion in the absence of normal osmotic (hypertonicity) or hemodynamic (reduced effective arterial blood volume) stimuli. 

Response: Thank you for your remark. We have implemented it.

Line 145: “fluid retention” should be “water retention” 

Response: Thank you for your remark. We have implemented it.

Line 156: “ionic” should be “electrolyte” 

Response: Thank you for your remark. We have implemented it.

Line 157: "accompanying" should be “associated with”

Response: Thank you for your remark. We have implemented it.

Line 159 -160: “allowing the administration of acyclovir as empiric treatment”. I am unclear what you mean by this line. Would rewrite. 

Response: Thank you for your remark. We have rewritten this sentence.

Line 172-175: I would include more explanation of the population studied: A cross sectional study of adults with CNS infection found that hyponatremia was present in 39% of cases.  Also I do not see where you got the “two-thirds of microorganisms identified from CSF were viruses”. I don't see the ⅔ reference. 

Response: Thank you very much for this insightful comment. We have revised the manuscript to provide a clearer description of the study population and to specify the source of the “two-thirds” statement. The study by Lim et al. was a cross-sectional analysis of 184 patients with clinical evidence of CNS infection and abnormal cerebrospinal fluid composition. A specific infectious organism was identified in 54% of patients, and—as reported in Section 3.2 (“Microbiology”) of the article (Table 2)—two-thirds of these were viruses, most commonly enteroviruses, followed by HSV and VZV. We have now clarified this point in the text to avoid ambiguity.

Line 187: put “a” before “CNS infection” 

Response: Thank you for your remark. We have implemented it.

Line 203: “are” should be “is”

Response: Thank you for this remark. We have implemented it.

Line 234-235: This line is a bit confusing since at first you say as symptoms improve but then you say the lack of improvement?  I would rewrite this sentence and would recommend “resolve” instead of “withdraw”.  “As the symptoms generally withdraw with the gradual equilibration of sodium, the lack of improvement, when normonatremia is achieved should prompt a search for other causes.”

Response: Thank you for this remark. We have implemented it.

Round 2

Reviewer 1 Report

Comments and Suggestions for Authors

The authors have addressed my concerns. The case novelty is now clear, as well as the conclusions.